# Vitamin D Receptor Expression Limits the Angiogenic and Inflammatory Properties of Retinal Endothelial Cells

**DOI:** 10.3390/cells12020335

**Published:** 2023-01-16

**Authors:** Yong-Seok Song, Nasim Jamali, Christine M. Sorenson, Nader Sheibani

**Affiliations:** 1Departments of Ophthalmology and Visual Sciences, University of Wisconsin School of Medicine and Public Health, Madison, WI 53705, USA; 2McPherson Eye Research Institute, University of Wisconsin School of Medicine and Public Health, Madison, WI 53705, USA; 3Department of Pediatrics, University of Wisconsin School of Medicine and Public Health, Madison, WI 53705, USA; 4Department of Cell and Regenerative Biology, University of Wisconsin School of Medicine and Public Health, Madison, WI 53705, USA; 5Department of Biomedical Engineering, University of Wisconsin-Madison, Madison, WI 53705, USA

**Keywords:** vitamin D, retinal vasculature, inflammation, STAT1, eNOS, NF-κB, Piezo1, Piezo2

## Abstract

The integrity of retinal endothelial cell (EC) is essential for establishing and maintaining the retinal blood barrier to ensure proper vision. Vitamin D is a hormone with known protective roles in EC function. The majority of vitamin D action is mediated through the vitamin D receptor (VDR). VDR is a nuclear receptor whose engagement by vitamin D impacts the expression of many genes with important roles in regulation of angiogenesis and inflammation. Although many studies have investigated vitamin D-VDR action in cardiovascular protection and tumor angiogenesis, its impact on retinal EC function and regulation of ocular angiogenesis and inflammation is exceedingly limited. We previously showed calcitriol, the active form of vitamin D, is a potent inhibitor of retinal neovascularization in vivo and retinal EC capillary morphogenesis in vitro. Here, using retinal EC prepared from wild-type (*Vdr*^+/+^) and VDR-deficient (*Vdr*^−/−^) mice, we show that retinal EC express VDR and its expression is induced by calcitriol. The lack of VDR expression had a significant impact on endothelial cell–cell and cell–matrix interactions. *Vdr*^−/−^ retinal EC proliferated at a slower rate and were more adherent and less migratory. They also exhibited increased expression levels of inflammatory markers driven in part by sustained activation of STAT1 and NF-κB pathways and were more sensitive to oxidative challenge. These changes were attributed, in part, to down-regulation of endothelial nitric oxide synthetase, enhanced hepcidin expression, and increased intracellular iron levels. Taken together, our results indicate that VDR expression plays a fundamental role in maintaining the proper angiogenic and inflammatory state of retinal EC.

## 1. Introduction

Vitamin D is a hormone with important roles in calcium homeostasis, immune system regulation, cell growth, and survival providing cardiovascular protection and anti-cancer activities [1]. Many studies have attributed these vitamin D activities to their ability to modulate inflammatory and angiogenesis processes. Epidemiological studies indicate that vitamin D insufficiency is associated with hypertension and endothelial dysfunction [2,3]. Low vitamin D also increases the risk of Alzheimer’s disease and vascular malfunction [4,5]. We previously showed that calcitriol, the active form of vitamin D, is effective in mitigating retinal neovascularization in a preclinical mouse model of retinopathy of prematurity in vivo [6]. Although calcitriol did not significantly affect the proliferation and migration of retinal endothelial cells (EC) in vitro, it did block their ability to undergo capillary morphogenesis, consistent with calcitriol’s ability to mitigate retinal neovascularization in vivo. However, the detailed mechanisms responsible for antiangiogenic activity of vitamin D, and the role vitamin D receptor (VDR) plays in retinal neurovasculature homeostasis are incomplete.

The major action of vitamin D is mediated through its receptor (VDR), a nuclear receptor with transcription regulatory activities [1]. The impact of vitamin D and VDR action on EC from various tissues have been the subject of numerous studies. Vitamin D exhibits anti-inflammatory effects on human coronary arterial EC [7], and mediates its antiproliferative effects in transformed EC, at least in part, through the down-regulation of NF-κB and P38 MAP kinase pathways and nitric oxide (NO) production [8]. Vitamin D also protects EC by inhibiting the expression of tissue factor and inflammatory cell adhesion molecules through modulation of these signaling pathways [9], and mitigates the loss of EC biomarkers expression including PECAM-1 and VE-cadherin [10]. In this way, vitamin D prohibits blood–brain barrier disruption mediated by hypoxia/reoxygenation [11]. Vitamin D also protects EC from oxidative stress by suppressing superoxide anion production, maintaining mitochondrial function and cell viability, engaging survival kinases, and inducing NO production [12]. It also inhibits endoplasmic reticulum stress [13] and reduces oxidative stress by enhancing glutathione production. In addition, vitamin D deficiency in healthy individuals is associated with increased monocyte-platelet aggregates and monocyte-EC adhesion contributing to the proinflammatory state associated with atherosclerosis [3]. However, the contribution of VDR to these activities in retinal vascular cells needs investigation.

To further delineate the molecular and cellular mechanisms responsible for antiangiogenic activity of vitamin D in the retina we used the *Vdr*^−/−^ mice. We showed that the antiangiogenic activity of calcitriol, in a mouse model of oxygen-induced ischemic retinopathy, was VDR dependent [14]. In addition, we showed retinal pericytes (PC) express VDR, and its expression is induced by calcitriol. Calcitriol inhibited the proliferation and migration of retinal PC in culture [15]. Furthermore, using retinal PC prepared from *Vdr*^−/−^ mice, we showed although the inhibition of retinal PC proliferation was VDR dependent, the inhibition of their migration by calcitriol was not [15]. We also demonstrated the important role of vitamin D signaling in enhanced production of VEGF in PC incubated with calcitriol, which derives the heterodimerization of PDGFRβ and VEGFR2 on PC mitigating signaling through either receptor [15]. This was previously proposed as a potential mechanism for turning off proangiogenic activity of PC during blood vessel development [16]. However, the cell autonomous role VDR expression plays in modulation of retinal EC function remains unexplored.

Vitamin D is shown to promote the production of endothelin and NO in cultured EC through VDR [17], and modulate arterial stiffness [2]. The elimination of VDR in vascular EC impacts their function, likely as a result of reduced NO production, contributing to cardiovascular disease and hypertension [18]. The activation of VDR in EC also results in increased VEGF and CuZn-SOD expression [19]. In addition, reduced VDR levels in circulating progenitor cells enhances the risk for coronary artery disease [20]. Furthermore, VDR knockdown in EC results in their activation and increased VCAM-1, ICAM-1, and IL-6 expression. These changes enhance the adhesive interactions between EC and inflammatory cells, and activate the NF-κB and MAP kinase signaling pathways [21]. Vitamin D-VDR axis is now recognized as a key signaling pathway in suppressing NF-κB pathway. Thus, appropriate vitamin D levels and adequate VDR expression in vascular cells could be critical for angiogenic and oxidative defense functions.

To address the cell-autonomous impact of VDR expression on retinal EC function, we used retinal EC prepared from *Vdr*^+/+^ and *Vdr*^−/−^ mice. We showed that lack of VDR expression has a significant impact on angiogenic and inflammatory properties of retinal EC, which are significantly impacted by the modulation of intracellular iron levels and cellular redox homeostasis.

## 2. Materials and Methods

### 2.1. Experimental Animals

All experiments were carried out in accordance with the Association for Research in Vision and Ophthalmology Statement for the Use of Animals in Ophthalmic and Vision Research and were approved by the Institutional Animal Care and Use Committee of the University of Wisconsin School of Medicine and Public Health. Immorto mice expressing a temperature-sensitive SV40 large T antigen were obtained from Charles River Laboratories (Wilmington, MA, USA). The vitamin D receptor-deficient (*Vdr*^−/−^) mice (B6.129S4-Vdrtm1Mbd/J; stock#: 006133), were obtained from Jackson Laboratories (Bar Harbor, ME, USA). *Vdr*^−/−^ mice were crossed with Immorto mice and the immorto; *Vdr*^−/−^ mice (all on C57BL/6 background) were genotyped by polymerase chain reaction (PCR) analysis of DNA extracted from tail tips. The screening primers used for genotyping were: Vdr mutant: 5′-CACGAGACTAGTGAGA CGTG-3′, Vdr wild type: 5′-CTCCATCCCCATGTGT CTTT-3′; and Vdr common: 5′-TTCTTCAGTGGCCA GCTCTT-3′, as suggested by the supplier. Mice were also screened for the presence of Rd1 and Rd8 mutations, which were absent.

### 2.2. Isolation and Culture of Vdr^−/−^ Retinal EC

Eyes from on litter (6 to 10 pups) of 4-week-old *Vdr*^+/+^ and *Vdr*^−/−^ immorto mice by collection of the retinas under a dissecting microscope. Twelve to fourteen retinas (from one litter) were pooled together, washed with serum-free Dulbecco’s modified Eagle’s medium (DMEM, D-5523, Sigma, St. Louis, MO, USA), cut into small pieces with a sterile razor blade in a 60 mm tissue culture dish, and digested with 2 mL of collagenase type I (1 mg/mL in serum-free DMEM, LS004194; Worthington, Lakewood, NJ, USA) at 37 °C for 45 min. Following the digestion, the digested retinas were washed with 5 mL of DMEM containing 10% fetal bovine serum (FBS, 26140-079; Gibco, Grand Island, NY, USA), centrifuged for 5 min at 400× *g*, resuspended in 5 mL of DMEM with 10% FBS, and filtrated through a double layer of sterile 40 μm nylon mesh (Sefar America Inc., Fisher Scientific, Hanover Park, IL, USA). The filtrate was centrifuged at 400× *g* for 10 min and washed twice with DMEM with 10% FBS. The pellet was suspended in 1 mL of DMEM containing 10% FBS and incubated with sheep anti-rat magnetic beads (11035; Invitrogen, Waltham, MA, USA) pre-coated with anti-platelet endothelial cell adhesion molecule-1 (PECAM-1) antibody (MEC 13.3, 553370; BD Bioscience, Bedford, MA, USA) at 4 °C on a rocker for 1.5 h. After affinity binding, magnetic beads were collected using a magnetic tube holder and rinsed six times with DMEM with 10% FBS. The bound cells were plated in a single well of a 24-well plate precoated with fibronectin (2 μg/mL prepared in serum-free DMEM, 354008; Corning, Steuben, NY, USA). The endothelial cells were cultured in DMEM containing 10% FBS, 2 mM L-glutamine (25030-081; Gibco), 2 mM sodium pyruvate (11360-070; Gibco), 20 mM HEPES (15630-080; Gibco), 1% non-essential amino acids (11140-050; Gibco), 100 μg/mL streptomycin, 100 U/mL penicillin (15140-122; Gibco), 55 U/mL heparin (H3149; Sigma), 100 μg/mL endothelial growth supplement (E2759; Sigma), and 44 U/mL murine recombinant interferon-γ (485-MI-100, R&D Systems, Minneapolis, MN, USA). Cells were maintained at 33 °C with 5% CO_2_ and progressively passed to larger plates and maintained on 1% gelatin (G1890; Sigma)-coated 60 mm tissue culture plates (12556001; Thermo Fisher).

### 2.3. RNA Purification and Real Time Quantitative PCR (qPCR) Analysis

The total RNA from retinal EC was extracted using a combination of TRIzol reagent (15596026; Invitrogen) extraction and RNeasy mini kit (74104; Qiagen, Valencia, CA, USA) column purification. The cDNA synthesis was performed from 1 μg of total RNA using RNA to cDNA EcoDry Premix (Double Primed) kit (639549; Clontech, Mountain View, CA, USA). 10-fold serial dilutions of cDNA were used as templates in qPCR assays, performed in triplicate on Mastercycler Realplex (Eppendorf, Enfield, CT, USA) using the TB-Green Advantage qPCR Premix (639676; Clontech). Standard curves were prepared from known quantities for standard target gene with linearized plasmid DNA. The linear regression line for DNA was determined from relative fluorescent units (RFU) at a threshold fluorescence (Ct). Target genes from cell extracts were quantified by comparing the RFU at the Ct to the standard curve and normalized by the simultaneous amplification of Rpl13a, a housekeeping gene. The qPCR primer sequences are listed in Table 1.

### 2.4. Western Blot Analysis

Cells were washed in PBS and lysed in radioimmunoprecipitation assay buffer containing 50 mM HEPES pH 7.5, 100 mM NaCl, 0.1 mM EDTA, 1 mM CaCl_2_, 1 mM MgCl_2_, 1% Triton X-100, 1% NP-40, 0.5% deoxycholate, and protease inhibitor cocktail (11836153001; Roche Biochemicals, Mannheim, Germany), briefly sonicated, and centrifuged at 400× *g* for 30 min at 4 °C. BCA protein assay kit (23225; Thermo Fisher) was used to determine protein concentrations. A total of 25–30 μg of protein samples were mixed with appropriate volumes of 6x SDS-sample buffer and separated by sodium dodecyl sulfate-polyacrylamide gel electrophoresis (4–20% Tris-glycine gels, Invitrogen) and transferred to a nitrocellulose membrane. Membranes were blocked with blocking buffer (5% nonfat dry milk and 0.05% Tween-20 in TBS) for 1 h at room temperature and incubated with primary antibodies diluted to 1:1000 in blocking buffer at 4 °C overnight. The membranes were washed and incubated with horseradish-peroxidase conjugated secondary antibody for 1 h at room temperature. Bands were detected using a chemiluminescence substrate (Amersham ECL Western Blotting Detection Reagent, 10600001; Cytiva, Marlborough, MA, USA). Antibodies used were as follows: anti-VDR (SC-13133; Santa Cruz, Dallas, TX), anti-ZO-1 (MABT11; Millipore), anti-VE-cadherin (550548; BD Bioscience), anti-N-cadherin (610920; BD Bioscience), anti-p120 Catenin (610133; BD Bioscience), anti-β-catenin (610154; BD Bioscience), anti-Fibronectin (F3648; Sigma), anti-Osteopontin (AF808, R&D), anti-SPARC (AF942; R&D), anti-TSP1 (MS-421P; Neomarker), anti-TSP2 (611150; BD Bioscience), anti-Phospho NF-ĸB P65 (3033; Cell Signaling, Danvers, MA, USA), anti- NF-ĸB P65 (8242; Cell Signaling), anti-Phospho STAT1 (9172; Cell Signaling), anti-STAT1 (9134; Cell Signaling), anti-eNOS (SC-654; Santa Cruz) and BMP6 (ab155693; Abcam, Waltham, MA, USA). Blots were stripped and incubated with anti-β-actin (MA5-157391; Invitrogen) antibody for loading control. The band intensities were measured and normalized against β-actin using ImageJ version 1.52J (National Institute of Health, Bethesda, MD, USA).

### 2.5. Flow Cytometry

Plates of retinal EC were rinsed with phosphate-buffered saline (PBS, D1408; Sigma) containing 0.04% EDTA and incubated with 2 mL of cell dissociation solution (TBS: Tris buffered saline; 20 mM Tris-HCL and 150 mM NaCl; pH 7.6) containing 2 mM EDTA and 0.05% bovine serum albumin (BSA, BP9703-100, Thermo Fisher). The cells were collected with DMEM containing 10% FBS, washed once with 5 mL TBS, fixed in 0.5 mL of 2% paraformaldehyde for 30 min on ice, and blocked in 1% goat serum for 20 min on ice. The cells were pelleted, resuspended in 0.5 of TBS with 0.1% BSA containing appropriate dilution of primary antibody (as recommended by the supplier), and incubated for 40 min on ice. The primary antibodies used were rat anti-PECAM-1 (553370; BD Bioscience) and rat anti-VE-cadherin (550548; BD Bioscience). Cells were then rinsed twice with TBS and incubated with the appropriate FITC-conjugated secondary antibody for 30 min on ice. Then, cells were washed twice with TBS and resuspended in 0.5 mL TBS with 1% BSA and analyzed by a FACScan caliber flow cytometer (Becton-Dickinson, Franklin Lakes, NJ, USA). Ten thousand cells were analyzed for each sample and analysis was performed by using FlowJo software (FLOWJO, LLC, Ashland, OR, USA, version 10). Cells incubated with secondary antibodies without primary antibodies were used as controls. Relative median fluorescence intensities (MFI) were determined by normalizing MFI of targets relative to MFI of the controls.

### 2.6. Indirect Immunofluorescence Staining

Cells (1 × 10^4^) were plated on fibronectin-coated 4-well chamber slides (PEZGS0416; Millipore, 5 µg/mL in serum-free DME). To determine localization of cell–cell junction molecules, 5 × 10^4^ cells were seeded and allowed to reach confluence. Cells were gently washed with PBS, fixed with cold 4% PFA for 10 min on ice, permeabilized with 0.1% Triton-X100 in PBS for 15 min at room temperature, and blocked with 1% BSA in TBS for 30 min. Slides were washed with TBS and incubated with primary antibodies (1:400) in 1% BSA in TBS at 4 °C overnight. Antibodies were used as follows: anti-VE-Cadherin (BD Bioscience), β-catenin (BD Biosciences), ZO-1 (Invitrogen), Vinculin (V4505, Sigma), and FITC-conjugated phalloidin (Sigma). The cells were rinsed with TBS and incubated with appropriate fluorescent-dye conjugated secondary antibodies (1:1000, Jackson ImmunoResearch) for 1 h at room temperature. The cells were then rinsed with TBS and incubated with DAPI (1:2000, D1306; Invitrogen) for 5 min and mounted on glass slides using a Fluoromount-G mounting solution (0100-01; SouthernBiotech, Birmingham, AL, USA). The cells were photographed with a Zeiss Fluorescence microscope (Axiophot, Zeiss, Germany) equipped with a digital camera.

### 2.7. Scratch Wound Assays

Cells (6 × 10^5^) were plated in 60 mm tissue culture plates and allowed to achieve confluence (2–3 days). Confluent cell monolayers were wounded with sterile plastic 200-μL micropipette tips, washed two times with DMEM containing 10% FBS, and fed with EC growth medium. Wound closure was monitored and photographed at 0, 24, and 48 h in digital format. For quantitative assessment, The wound areas were measured using ImageJ software with the MRI Wound Healing plugin [22].

### 2.8. Transwell Migration Assays

The bottoms of Costar trans wells with 8 μm pore size (3422; Corning) were coated with fibronectin (2 μg/mL in PBS, Corning) overnight at 4 °C. Next day, the bottom side of the transwell was washed with PBS and blocked with 2% BSA in PBS for 1 h at room temperature. Retinal EC (1 × 10^5^) were added to the top of the transwell membrane, and the cells were incubated for 4 h at 33 °C. The cells that had migrated through the membrane were fixed with 4% paraformaldehyde (PFA) for 10 min at room temperature and stained with hematoxylin/eosin. The membranes were mounted on a glass slide and the mean number of cells that migrated through the filter was determined by counting ten high power fields (×200).

### 2.9. Cell Adhesion Assays

A total of 96 well flat-bottom plates (Nunc Immunoplate Maxisorp, Fisher Scientific) were coated with various concentrations of extracellular matrix proteins, including fibronectin (Corning), vitronectin (2349-VN-100; R&D), collagen I (354243; Corning), and collagen IV (354233; BD Biosciences), or BSA as control. The proteins were serially diluted in TBS containing 2 mM CaCl_2_ and MgCl_2_ (TBS with Ca/Mg) and coated (50 μL) the wells of the 96 well plates overnight at 4 °C. Wells were rinsed with 200 μL of TBS with Ca/Mg and blocked with TBS Ca/Mg containing 1% BSA (200 μL/well) at room temperature for 1 h. Cells were dissociated with cell dissociation buffer (3 mL/60 mm tissue culture plate, 2 mM EDTA and 0.05% BSA in TBS), washed with TBS, and resuspended in cell binding buffer (20 mM HEPES, pH 7.4, 150 mM NaCl, 4 mg/mL BSA) at approximately 6 ×10^5^ cells/mL. The plates were rinsed with TBS with Ca/Mg and incubated with 3×10^4^ cells (50 μL of TBS with Ca/MG and 50 of cell suspension) at 37 °C for 2 h. After incubation, the plates were gently washed with 200 μL of TBS with Ca/Mg to remove non-adherent cells until no cells were left in the wells coated with BSA. The adherent cells were lysed with 100 μL of lysis buffer (50 mM sodium acetate pH 5.0, 1% Triton X-100, 6 mg/mL p-nitrophenyl phosphate) and incubated at 4 °C overnight. The reaction was neutralized by adding 50 μL 1 M NaOH and the absorbance was measured at 405 nm using a microplate reader (Thermomax, Molecular Devices, Sunnyvale, CA, USA). All samples were done in triplicates and repeated twice.

### 2.10. Cell Proliferation Assays

Cells (2 × 10^4^) were plated on multiple sets of gelatin-coated 60 mm tissue culture plates and counted every other day for two weeks. Cells were fed on the days not counted. The mean cell numbers were determined using a hemocytometer. All samples were conducted in triplicates and repeated twice.

### 2.11. Cell Viability Assay

A total of 3×10^3^ cells were plated on 96-well plates (130188, Thermo Fisher). The next day, the cells were incubated with different concentrations of H_2_O_2_ (0–4 mM) for 48 h and further incubated with MTS solution (G5421; Promega, Madison, WI, USA) for 2 h. The viability of cells was accessed by measuring absorbance at 490 nm using a plate reader (Thermomax, Molecular Devices) and determined as a percentage of control (untreated) cells.

### 2.12. Measuring Intracellular Iron Levels

A total of 2.0 × 10^4^ cells were plated on fibronectin-coated 4-well chamber slides (2 µg/mL in serum-free DMEM). The next day, the cells were rinsed with serum-free medium three times and incubated with 1 µM FerroOrange (F374-10; Dojindo, Rockville, MD, USA) for 30 min in a 37 °C incubator. The cells were photographed with an inverted fluorescence microscope (EVOS FL Digital Inverted Fluorescence microscope (Invitrogen). For quantitative assessment of the data, the mean integrated fluorescence intensities were determined from at least 6 images from different locations of the slides using ImageJ. Alternatively, 3.0 × 10^4^ cells were plated on 96-well plates and stained with FerroOrange as described above. The fluorescence levels were measured by a fluorescence plate reader (SpectraMax i3x Multi-mode Microplate Reader, Molecular devices) at 543/580 nm and normalized to total protein levels determined by BCA protein assay (ThermoFisher).

### 2.13. Statistical Analysis

Quantitative results were expressed as mean ± SD. Statistical differences were evaluated with the student’s unpaired t-test. All data analysis was performed with GraphPad Prism version 8 (GraphPad Software, La Jolla, CA, USA). Statistical analyses *p* values ≤ 0.05 were considered significant.

## 3. Results

### 3.1. Isolation and Characterization of Vdr^−/−^ Retinal EC

Retinal EC were prepared from *Vdr*^+/+^ and *Vdr*^−/−^ immorto mice using a method previously reported by our group [23] and detailed in Material and Methods. At least three different isolations of these cells were used in all experiments. The absence of VDR expression was verified by qRT-PCR and Western blot analysis. Cell morphology was assessed at sub-confluent and confluent states using light microscopy (Figure 1A). While *Vdr*^+/+^ EC exhibited a uniform and closely packed morphology typical of EC, *Vdr*^−/−^ retinal EC showed a more elongated and spindly morphology with limited cell–cell aposition. *Vdr*^−/−^ retinal EC failed to form a uniform and closely apposed monolayers compared with *Vdr*^+/+^ retinal EC. The qPCR and Western blot analysis showed that *Vdr*^−/−^ retinal EC lacked VDR expression and, unlike the *Vdr*^+/+^ cells which increase VDR expression in response to calcitriol (10 µM, 24 h) treatment, *Vdr*^−/−^ cell as expected did not show a response (Figure 1B,C). These results are consistent with our previous studies with retinal PC which express significantly higher levels of VDR and it was induced by calcitriol [15]. We next determined the expression of EC markers by flow cytometry analysis. Retinal EC expressed VE-cadherin and PECAM-1 on their surface (Figure 1D). However, PECAM-1 levels were lower in *Vdr*^−/−^ retinal EC, consistent with a previous study demonstrating increased PECAM-1 levels in heart EC following activation of VDR by calcitriol [10]. Collectively, these results suggested potential changes in cell–cell interactions and the integrity of adherens and tight junctions, which play important roles in proper vascular function [24].

We next examined the localization and expression levels of cell–cell junction molecules by Western blot analysis and indirect immunofluorescence staining of *Vdr*^+/+^ and *Vdr*^−/−^ retinal EC. Adherens junctions are composed of the transmembrane cadherins and intracellular proteins such as p120-catenin and β-catenin. EC express cadherins including VE-cadherin and N-cadherin [25]. The *Vdr*^+/+^ retinal EC showed well-organized junctional localization and significant expression of VE-cadherin (Figure 2A,B). In contrast, *Vdr*^−/−^ retinal EC lacked junctional localization of VE-cadherin, whose expression was also greatly reduced. However, the expression of N-cadherin, p120 catenin, and β-catenin was increased in *Vdr*^−/−^ retinal EC. ZO-1 is a peripheral membrane protein that plays important roles in assembly of both adherens and tight junctions by interacting with other transmembrane and cytoplasmic proteins [26]. *Vdr*^−/−^ retinal EC showed up-regulation of ZO-1 expression that lacked junctional localization (Figure 2A–C). These results support a role for VDR expression in the regulation of cell–cell junction in retinal EC. This is consistent with the changes in the levels of these proteins and increased permeability noted in other types of EC with knockdown or deletion of VDR [21,27].

### 3.2. Vdr^−/−^ Retinal EC Were Less Migratory

Migration is one of the fundamental characteristics of EC during angiogenesis. To examine the migratory properties of retinal EC, we performed a scratch wound migration assay. Confluent monolayers of *Vdr*^+/+^ and *Vdr*^−/−^ retinal EC were wounded and closure was monitored every day. After 48 h, the wound in *Vdr*^+/+^ retinal EC was completely closed, but wound closure was significantly delayed in *Vdr*^−/−^ retinal EC (Figure 3A). Similar results were observed in transwell migration assays (Figure 3B). Focal adhesions and actin stress fibers play important roles in cell migration. We stained *Vdr*^+/+^ and *Vdr*^−/−^ retinal EC with anti-vinculin antibody (focal adhesion) and FITC-conjugated phalloidin (F-actin) to assess the appearance of focal adhesions and actin fibers. *Vdr*^−/−^ EC showed increased localization of actin stress fibers on the cell periphery with the formation of prominent focal adhesions when compared to *Vdr*^+/+^ retinal EC (Figure 3C), consistent with the reduced rate of cell migration noted in *Vdr*^−/−^ retinal EC.

### 3.3. Vdr^−/−^ Retinal EC Were More Adherent

The decreased cell migration in *Vdr*^−/−^ retinal EC suggested alteration in their adhesion properties. We next examined the adhesion of *Vdr*^+/+^ and *Vdr*^−/−^ retinal EC to various ECM proteins, including fibronectin (FN), vitronectin (VN), collagen I, and collagen IV. Figure 4 shows that *Vdr*^−/−^ retinal EC were more adherent on the tested ECM proteins compared with *Vdr*^+/+^ cells (Figure 4). These results were consistent with the reduced cell migration noted in *Vdr*^−/−^ retinal EC.

### 3.4. Altered Expression of ECM Proteins in Vdr^−/−^ Retinal EC

Endothelial cell adhesion to ECM proteins through specific cell surface integrins mediates adhesion and migration [28]. We investigated the expression of ECM proteins in *Vdr*^+/+^ and *Vdr*^−/−^ retinal EC by performing Western blot analysis of cell lysates (cell-associated proteins) and conditioned medium (secreted proteins). Figure 5A shows *Vdr*^−/−^ retinal EC produced higher levels of osteopontin, SPARC, and TSP2 but similar levels of fibronectin compared with *Vdr*^+/+^ cells. The *Vdr*^−/−^ retinal EC produced lower levels of TSP1. The quantification of data is shown in Figure 5B. These observations indicate that VDR expression affects adhesive and migratory properties of retinal EC impacted by changes in production of ECM proteins.

### 3.5. Increased Expression of Pro-Inflammatory Mediators and Activation of Proinflammatory Transcription Factors in Vdr^−/−^ Retinal EC

Osteopontin, SPARC, and TSP1 play important roles in inflammatory responses, such as immune cell migration and infiltration [29,30]. Thus, changes in these ECM protein levels may indicate alterations in the regulation of inflammatory mediators in *Vdr*^−/−^ retinal EC. To address this question, we performed qPCR analysis to compare inflammatory mediator expression levels in retinal EC. We observed increased expression of key proinflammatory mediators including MCP-1, IL-6, IL-33, and its receptor St2 in *Vdr*^−/−^ retinal EC compared with *Vdr*^+/+^ cells. We also noted increased expression of other genes with important roles in inflammatory process including Icam-1, Vcam-1, Ager (receptor for advanced glycation endproducts, RAGE), F3 (Coagulation factor III), Fas, Nos2 (iNOS), Vegf, Plau (Urokinase plasminogen activator, uPA), and Stat3 in *Vdr*^−/−^ retinal EC (Figure 6). Given the important roles of VE-cadherin and PECAM-1 as mechanosensors, whose expressions were affected by VDR expression, we next determined if the expression mechanosensing ion channels, namely Piezo1 and Piezo2 are affected. The expression of Piezo1, but not Piezo2, was detectable in *Vdr*^+/+^ retinal EC. However, the expression of both Piezo1 and Piezo2 was significantly upregulated in *Vdr*^−/−^ retinal EC (Figure 6). These results are consistent with changes in adhesive and migratory properties of these cells through alterations and remodeling of ECM proteins.

Signaling pathways of pro-inflammatory cytokines are mediated by transcription factors such as NF-κB. The activation of NF-κB is controlled by multiple cellular events, resulting in the phosphorylation of the p65 subunit. In order to determine the effect of VDR expression on NF-κB activity, the level of phosphorylated p65 was assessed by Western blot analysis (Figure 7A). We observed an increase in p65 phosphorylation in *Vdr*^−/−^ retinal EC compared to *Vdr*^+/+^ cells. In addition, we performed an immunofluorescence assay to detect NF-κB nuclear translocation in EC. The nuclear localization of phosphorylated p65 was noticeably increased in *Vdr*^−/−^ retinal EC (Figure 7B). We also observed increased phosphorylation of STAT1, an important transcription factor that mediates pro-inflammatory signaling pathways, in *Vdr*^−/−^ retinal EC (Figure 7C). Collectively, these results indicate that the absence of VDR expression is associated with a pro-inflammatory state of retinal EC.

### 3.6. Altered Cell Proliferation and Increased Oxidative Stress in Vdr^−/−^ Retinal EC

Pro-inflammatory mediators and their signaling pathways regulate a range of cellular events including cell proliferation, oxidative stress, and apoptosis. To determine whether lack of VDR expression impacts cell proliferation, we counted the number of cells during a 2-week period in culture. *Vdr*^−/−^ retinal EC showed a significant decrease in the rate of proliferation compared with *Vdr*^+/+^ cells (Figure 8A).

We next determined the sensitivity to oxidative stress by evaluating cell viability in retinal EC challenged with H_2_O_2_. Cells were exposed to different concentrations of H_2_O_2_ (1, 2, and 3 mM) for 2 days and cell viability was assessed using the MTS assay. The incubation with 3 mM H_2_O_2_ decreased the viability of *Vdr*^+/+^ retinal EC by 20%, while that of *Vdr*^−/−^ retinal EC was decreased by 50% (Figure 8B). These results indicate VDR expression is associated with regulation of redox homeostasis in retinal EC. To address this question, we determined the expression of eNOS, an enzyme with key roles in production of nitric oxide (NO) whose perturbation could drive redox dysregulation [31]. NO plays a key role in regulating EC homeostasis and is recognized as an early target in pathophysiology of several vascular diseases [32,33]. It also has an antioxidant property and can scavenge lipid peroxyl radicals [34]. We observed eNOS expression was significantly decreased in *Vdr*^−/−^ retinal EC (Figure 8C). These results demonstrated a significant role for VDR expression in the regulation of eNOS expression and is consistent with the reported impaired expression of eNOS in the aorta from endothelial cell-specific VDR knockout mice [18].

### 3.7. Increased Expression of BMP6 and Its Down-Stream Target Gene Hamp in Vdr^−/−^ Retinal EC

The bone morphogenetic protein (BMP) family has multiple roles throughout the body, and among them, alteration in BMP6 signaling is linked to EC dysfunction. BMP6 treatment of EC results in increased osteopontin expression [35] and disruption of VE-cadherin membrane localization [36], which were also observed in *Vdr*^−/−^ retinal EC (Figure 2 and Figure 5). These data suggest a potential alteration in BMP6 signaling in the absence of VDR expression in retinal EC. To further address the consequence of these changes, we examined BMP6 expression in retinal EC using qPCR and Western blot analysis. Here, we show *Vdr*^−/−^ retinal EC express significantly higher levels of BMP6 compared with *Vdr^+/+^* retinal EC (Figure 9A,B).

In the liver, sinusoidal endothelial cells express and secrete BMP6, acting on hepatocytes to enhance hepcidin expression [37]. The retina is separated from systemic circulation by the blood–retinal barrier, and the iron homeostasis in the retina is mainly modulated by local production of iron regulators including hepcidin [38]. Given that retinal EC expresses BMP receptors, increased BMP6 levels in *Vdr*^−/−^ retinal EC may enhance hepcidin expression in the retinal EC. qPCR analysis showed a significant increase in expression of hepcidin mRNA (Hamp) in *Vdr*^−/−^ retinal EC (Figure 9C). Hepcidin is a peptide that regulates iron homeostasis by blocking and enhancing the degradation of cellular iron exporter ferroportin, resulting in increased intracellular iron levels [39]. Thus, these results suggest that VDR expression is associated with the regulation of oxidative stress, at least in part, via maintaining iron homeostasis in retinal EC. This notion is supported by a study showing suppression of systemic hepcidin by vitamin D [40]. We also noted a significant increase in the amount of iron accumulation in *Vdr*^−/−^ retinal EC (Figure 9C).

## 4. Discussion

Vitamin D is a multifunctional hormone and, its active metabolite, 1,25-dihydroxy vitamin D3 (1,25(OH)_2_D_3_ or calcitriol), impacts various biological processes. Most, if not all, genomic actions of vitamin D are mediated by VDR, which belongs to the class II subfamily of the nuclear receptors. VDR acts as a ligand-inducible transcription factor that binds as a heterodimer with retinoid X receptor to vitamin D-responsive elements in the promoters of vitamin D target genes [41]. The control of calcium homeostasis and bone metabolism is considered the primary function of vitamin D and VDR. In addition, the concept that VDR is nearly ubiquitously expressed in all tissues and various types of cells indicates that VDR is involved in more functions than the above-mentioned classical roles [42]. Interestingly, a fully functional VDR evolved in a boneless vertebrate, and studies suggest that *VDR* was first specialized in the regulation of innate and adaptive immunity before it took the role in the modulation of calcium homeostasis [43].

Vitamin D deficiency has been linked to cardiovascular diseases and dysfunction of vascular smooth muscle cells and vascular endothelium [18,44]. Many tissues in the eye can activate and respond to vitamin D [15,45,46]. Epidemiological studies have established an association between levels of vitamin D and a wide range of ocular pathologies including myopia, AMD, diabetic retinopathy, and uveitis [45]. At the cellular level, vitamin D treatment reduces inflammation, enhances barrier function, and inhibits angiogenesis, thus providing a protective role in ocular health. Our previous studies showed that calcitriol inhibited ischemia-mediated retinal neovascularization, which was VDR-dependent [14]. VDR expression is also essential during normal mouse retinal vascular development [6,14]. Using retinal PC isolated from *Vdr*^+/+^ and *Vdr*^−/−^ mice, we previously reported that VDR expression has a significant impact on PC function [15]. However, its cell autonomous impact on retinal EC function remained unknown.

Here, we report the significant impact of VDR-deficiency on retinal EC proangiogenic and inflammatory characteristics, the majority of which are shared with previously reported changes in EC from other tissues where VDR was knocked down or deleted in EC. Knockdown of VDR in EC led to their activation characterized by increased expression of adhesion molecules ICAM-1 and VCAM-1, and inflammatory mediators including IL-6, concomitant with increased inflammatory cells adhesion and nuclear localization of NF-κB [11,18,20,21,47]. In addition, down-regulation of VDR promotes endothelial inflammatory responses in preeclampsia [48]. EC-targeted VDR null mice are impaired in EC-dependent vasorelaxation and exhibit an augmented blood pressure response to angiotensin II administration [18]. We observed lack of VDR expression had a significant effect on the morphology, cell–cell, and cell–matrix interactions of retinal EC impacting their proliferation, migration, inflammatory, and redox state. *Vdr*^−/−^ retinal EC proliferated at a slower rate, were less migratory, and more adherent on various ECM proteins evaluated here. These cells also exhibited enhanced inflammatory signaling, perhaps through sustained activation of NF-κB and STAT1 transcription factors, consistent with enhanced production of inflammatory mediators including IL-33.

IL-33 is recognized as a key immunomodulatory cytokine in many neurological diseases [49]. IL-33 induces proinflammatory, prothrombotic, and proangiogenic activation of microglia and EC [49,50,51,52,53]. It is also a crucial regulator of mast cell function, and as such might be an attractive target for treatment of allergic and inflammatory diseases [54,55]. IL-33 promotes angiogenesis, perhaps through enhanced IL-8 production [56]. IL-33 is recently shown to protect mice against hindlimb ischemic injury by enhancing EC proangiogenic activity [57]. In addition, deletion of IL-33 in EC blocks neovascularization during oxygen-induced ischemic retinopathy, while its incubation with retinal EC enhances their sprouting angiogenesis [58]. The increased production of tissue factor by IL-33 enhances the thrombotic properties of EC [59]. Thus, IL-33 signaling through its receptor ST2 plays a crucial role in EC activation and modulation of microglia-mediated neuroinflammation [50,58], which is likely impacted by the vitamin D-VDR signaling axis. The expression of inflammatory mediators including IL-33, Mcp-1, IL-6, Ager, Vegf, Plau, and Nos2, as well as the expression of ICAM-1, VCAM-1, St2, Stat3, F3, and Fas were significantly increased in *Vdr*^−/−^ retinal EC consistent with an inflammatory phenotype. No significant changes were noted in the expression of Cxcl1 and Cxcl2.

Our results indicate significant roles for VDR expression in modulation of cell–cell and cell–matrix interactions of retinal EC. We observed significant changes in the levels of VE-cadherin and ZO-1, as well as levels of ECM proteins with important regulatory roles in inflammation such as osteopontin, SPARC, and TSP1. The increased level of osteopontin in *Vdr*^−/−^ retinal EC is consistent with its reported role in enhanced vascular permeability, through disruption of cell–cell interactions, under diabetic conditions [60]. SPARC is another matricellular protein with an important role in inflammation, fibrosis, and angiogenesis [61]. Increased production of SPARC by retinal EC incubated with high glucose resulted in decreased migration, proliferation, and angiogenesis. SPARC also increased permeability by altering cell–cell interactions and increased collagen deposition [62,63]. In contrast, decreased expression of TSP1, a key ocular anti-inflammatory protein, also promotes inflammation [64]. Thus, alterations in expression of osteopontin, SPARC, and TSP1 may also contribute to alterations noted in *Vdr*^−/−^ retinal EC cell–cell and cell–matrix interactions, and inflammatory processes.

Endothelial cells are well recognized for their ability to sense mechanical forces such as shear stress and have the capacity to translate them into biochemical stimuli through mechanotransduction channels such as the mechanically activated cation channels Piezo1 and Piezo2 [65]. Piezo1 is essential for early vascular development and mediates flow-mediated increase in EC intracellular calcium [66,67]. Flow mediated activation of Piezo1 in EC mediates the activation of inflammatory signaling and changes in NO production [68,69]. Piezo2 modulates cellular cytoskeleton changes including actin-based stress fibers and distribution of focal adhesions through Rho A [70]. A recent study showed lung EC lacking Piezo2 expression were impaired in NO production, endothelial mesenchymal transition, and develop pulmonary hypertension [71]. Thus, changes in expression and activity of Piezo channels may contribute to the alterations in *Vdr*^−/−^ retinal EC noted here. The expression of these mechanically activated cation channels in the retinal EC have not been previously reported. We showed *Vdr*^+/+^ retinal EC express Piezo1, but not detectable amount of Piezo2. However, *Vdr*^−/−^ retinal EC expressed similar levels Piezo1 and Piezo2, which were significantly higher than their levels in *Vdr*^+/+^ retinal EC. The reason for the increased expression of Piezo1 and Piezo2 in *Vdr*^−/−^ retinal EC remains unknown but may suggest an important role for changes in cell–cell and cell–matrix interactions and inflammatory processes noted in these cells. In addition, the regulation of these channels through VDR-vitamin D axis and their role in regulation of calcium homeostasis deserves further investigation.

We also noted increased sensitivity to oxidative challenge in *Vdr*^−/−^ retinal EC suggesting increased oxidative stress in these cells. However, the underlying mechanisms and more specifically the role VDR plays in modulation of redox homeostasis is unclear. Iron is an essential element whose homeostasis is beneficial to the health of the tissue and organism. Systemic iron homeostasis is maintained by production of iron regulatory hormone hepcidin, produced by liver hepatocytes through their stimulation by BMP6 produced by liver sinusoidal endothelial cells [37,72]. Hepcidin regulates iron levels through modulation of ferroportin levels, the only iron exporter in vertebrates [73]. It is known that retinal EC play a key role in the regulation of local ocular iron homeostasis, given their role in retina–blood barrier function, likely by hepcidin both in a paracrine and/or autocrine fashion [74,75]. In chronic kidney disease, vitamin D deficiency was proposed to cause an increase in hepcidin production, which leads to anemia [40]. However, treatment with vitamin D suppressed hepcidin levels and increased ferroportin levels. These studies highlighted the important role of vitamin D in the regulation of hepcidin-ferroportin axis. Consistent with this notion, we noted a significant increase in hepcidin level and decreased level of ferroportin expression in *Vdr*^−/−^ retinal EC leading to increase intracellular iron accumulation. Increase intracellular iron level is known to lead to increased oxidative stress through the Fenton reaction generating hydroxyl radicals which are highly reactive and lead to excess lipid peroxidation, oxidative stress, and cellular damage [76]. Thus, regulation of hepcidin-ferroportin axis in retinal EC through VDR could play an important role in the regulation of ocular iron homeostasis and oxidative stress, and the vascular dysfunction associated with vitamin D deficiency.

In summary, these studies demonstrate an important role for VDR expression in modulation of angiogenic and inflammatory phenotype of retinal EC through modulation of cell–cell and cell–matrix interactions. The *Vdr*^−/−^ retinal EC were less migratory, more adherent, and less proliferative. They also produced proinflammatory ECM proteins and exhibited enhanced expression of various inflammatory mediators along with sustained activation of NF-κB pathway and expression of Stat1 and Stat3 transcription factors. These activities are consistent with noted increased sensitivity of these cells to oxidative challenge and increased oxidative stress. The increased oxidative stress is consistent with increased accumulation of iron in *Vdr*^−/−^ retinal EC mediated by enhanced expression of hepcidin and decreased levels of ferroportin. Thus, VDR activity is important in regulation of iron homeostasis in the retinal endothelium and in keeping inflammatory processes and cellular redox homeostasis in check.

## Figures and Tables

**Figure 1 cells-12-00335-f001:**
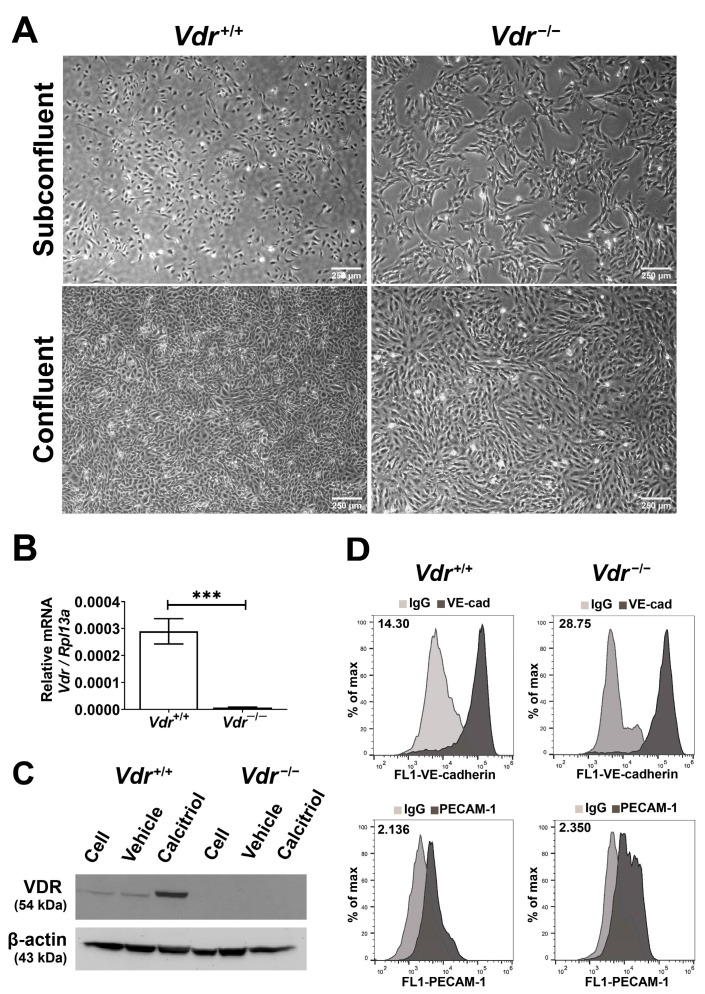
Isolation and characterization of mouse retinal EC. *Vdr*^+/+^ and *Vdr*^−/−^ retinal EC were prepared and cultured as described in Materials and Methods. (**A**) Morphology of cells at subconfluent and confluent. (**B**) Vitamin D receptor expression levels were accessed by qPCR analysis (n = 3, *** *p* < 0.001). (**C**) Cells were incubated with medium alone (cells), 0.41% ethanol (vehicle), or 10 µM calcitriol for 24 h and VDR levels were measured by Western blot analysis. (**D**) Expression levels of EC markers were determined by flow cytometry. VE-cad; VE-cadherin and PECAM-1; Platelet endothelial cell adhesion molecule-1. Relative median fluorescent intensities are indicated in the top left corner of each panel. These cells were also positive for B4-lectin (a mouse EC specific lectin) and lacked the expression of PDGFRβ and smooth muscle actin (pericyte markers; not shown).

**Figure 2 cells-12-00335-f002:**
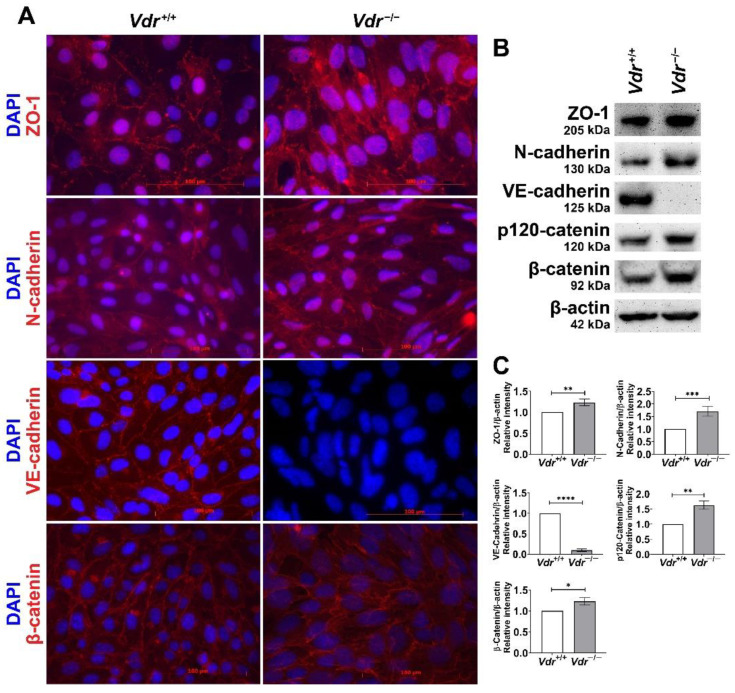
Cellular localization and expression levels of ZO-1, N-cadherin, VE-cadherin, p120-catenin, and β-catenin. (**A**) Retinal EC were cultured on fibronectin-coated chamber slides to confluence and stained with specific antibodies as described in Section 2. (**B**,**C**) Western blot analysis of junctional proteins (n = 3–4, * *p* < 0.05, ** *p* < 0.01, *** *p* < 0.001, **** *p* < 0.0001).

**Figure 3 cells-12-00335-f003:**
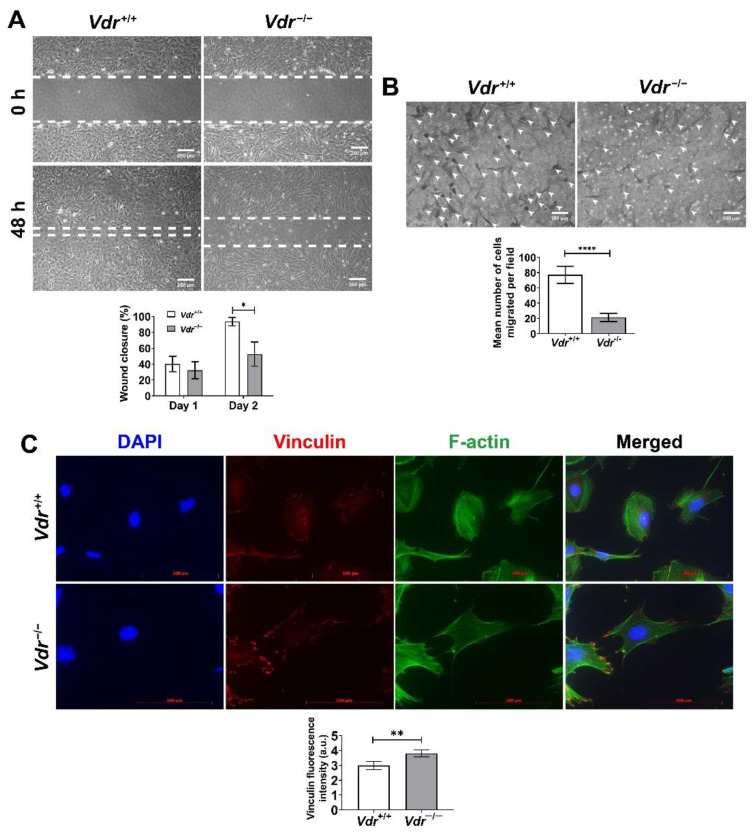
*Vdr*^−/−^ retinal EC were less migratory. The migration of *Vdr*+/+ and *Vdr*^−/−^ retinal EC were compared by scratch wound assay of retinal EC monolayers. Wound closure was monitored until 48 h. Using photographs of wounds (**A**) wound closure was quantitatively determined as described in Methods (n = 4, * *p* < 0.05). (**B**) Cell migration was also determined by performing a transwell migration assay. Cells were photographed (left panel) and the mean number of migrated cells perfield were counted for quantitative analysis (n = 6, **** *p* < 0.0001). Arrow heads point to cells migrated through the filter and adhered to the bottom of tranwell membrane (**C**) The indirect immunofluorescence staining using phalloidin (green; actin filaments), anti-vinculin antibody (red; focal adhesions), and DAPI (blue; nuclei). Vinculin levels were determined by measuring fluorescence intensities, as described in Methods (n = 4, ** *p* < 0.01).

**Figure 4 cells-12-00335-f004:**
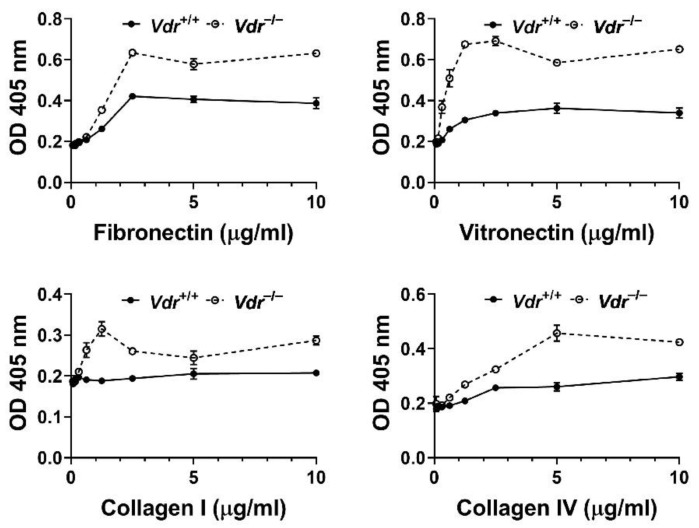
*Vdr*^−/−^ retinal EC were more adherent to extracellular matrix proteins. Adhesion of retinal EC to fibronectin, vitronectin, collagen I, and collagen IV was analyzed as described in Methods. OD; optical density. These experiments were repeated with two different isolations with similar results.

**Figure 5 cells-12-00335-f005:**
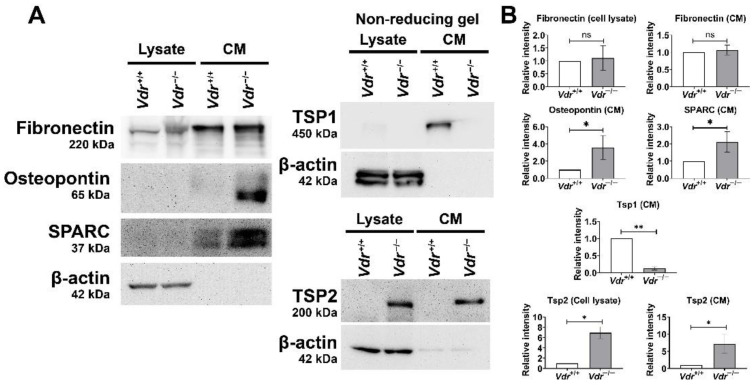
Altered expression of ECM proteins in *Vdr*^−/−^ retinal EC. Retinal EC were incubated with serum-free culture medium for 48 h. The collected cell lysates (Lysate) and conditioned medium (CM) were analyzed by Western blot analysis for fibronectin, osteopontin, SPARC, TSP1, and TSP2 with specific antibodies (**A**). The bands were quantified with densitometry and normalized by β-actin (**B**); n = 3–4, * *p* < 0.05, ** *p* < 0.01; ns: not significant).

**Figure 6 cells-12-00335-f006:**
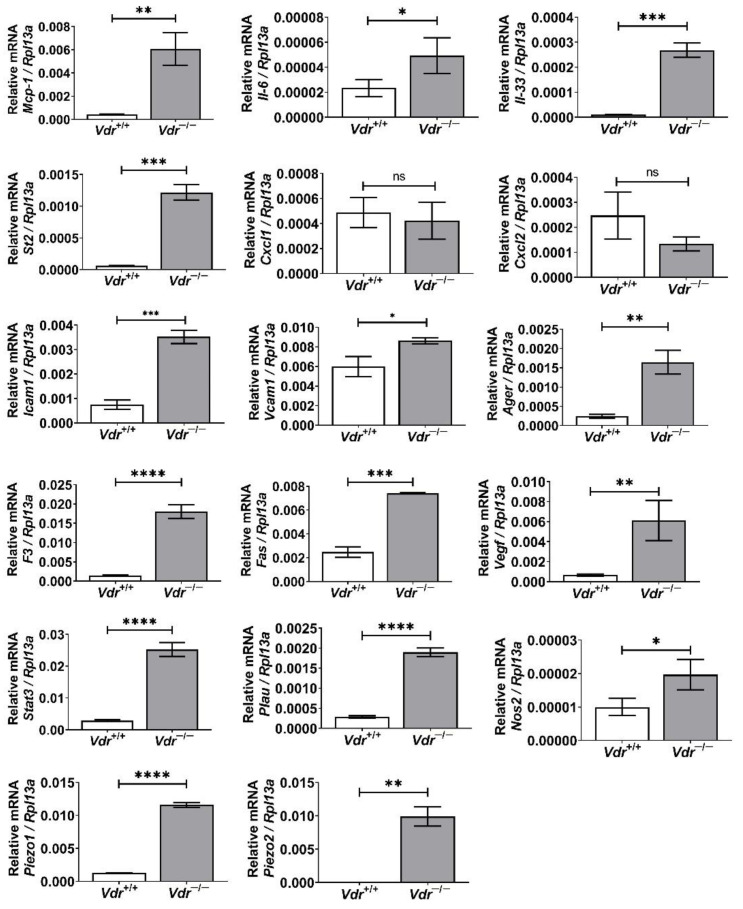
Increased expression of pro-inflammatory mediators in *Vdr*^−/−^ retinal EC. Expression levels of Mcp-1, Il-6, Il-33, St2, Cxcl1, Cxcl2, Icam-1, Vcam-1, Ager, F3, Fas, Vegf, Stat3, Plau, Nos2, Piezo1, and Piezo2 were determined by qPCR analysis. Expression levels of target genes were normalized by Rpl13a (60 S ribosomal protein L13a) expression. (n = 3, * *p* < 0.05, ** *p* < 0.01, *** *p* < 0.001, **** *p* < 0.0001; ns: not significant).

**Figure 7 cells-12-00335-f007:**
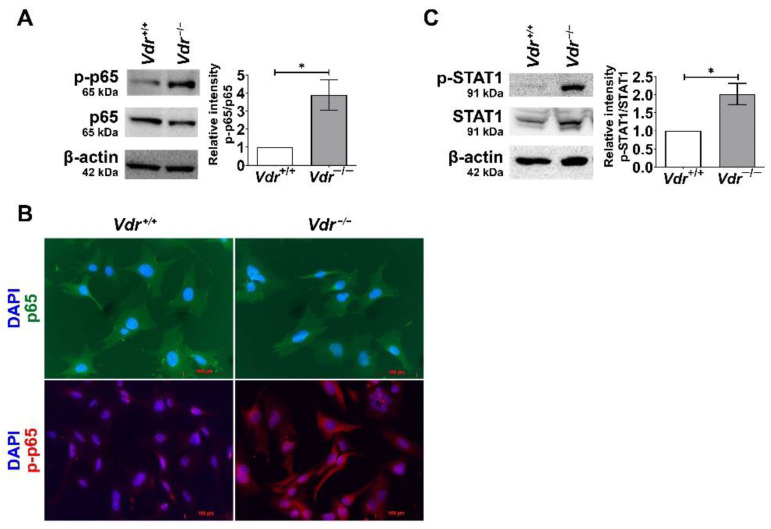
Upregulated activities of pro-inflammatory transcription factors in *Vdr*^−/−^ retinal EC. (**A**) phosphorylated NF-κB (p-p65) and NF-κB (p65) levels in *Vdr*^+/+^ and *Vdr*^−/−^ retinal EC were determined by Western blot analysis with specific antibodies. (n = 4, * *p* < 0.05) (**B**) Indirect immunofluorescent staining of p65 (green), p-p65 (red), and nuclei (blue) was performed to demonstrate localization of NF-κB. (**C**) Phosphorylated STAT1 and STAT1 levels in *Vdr*^+/+^ and *Vdr*^−/−^ retinal EC were determined by Western blot analysis with specific antibodies (n = 4, * *p* < 0.05).

**Figure 8 cells-12-00335-f008:**
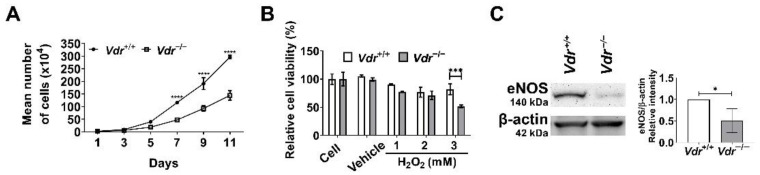
Altered cell proliferation and increased oxidative stress in *Vdr*^−/−^ retinal EC. (**A**) The rate of *Vdr*^+/+^ and *Vdr*^−/−^ retinal EC was accessed by counting the number of cells for 2 weeks as described in Materials and Methods (n = 3, **** *p* < 0.0001). (**B**) Sensitivity to oxidative stress induced by hydrogen peroxide (H_2_O_2_) in retinal EC was measured by MTS assay. *Vdr*^+/+^ and *Vdr*^−/−^ retinal EC were incubated with various concentrations of H_2_O_2_ for 2 days in 96-well plates and subjected to the MTS assay (n = 3, *** *p* < 0.001). (**C**) eNOS levels in *Vdr*^+/+^ and *Vdr*^−/−^ retinal EC were determined by Western blot analysis with a specific antibody (n = 4, * *p* < 0.05).

**Figure 9 cells-12-00335-f009:**
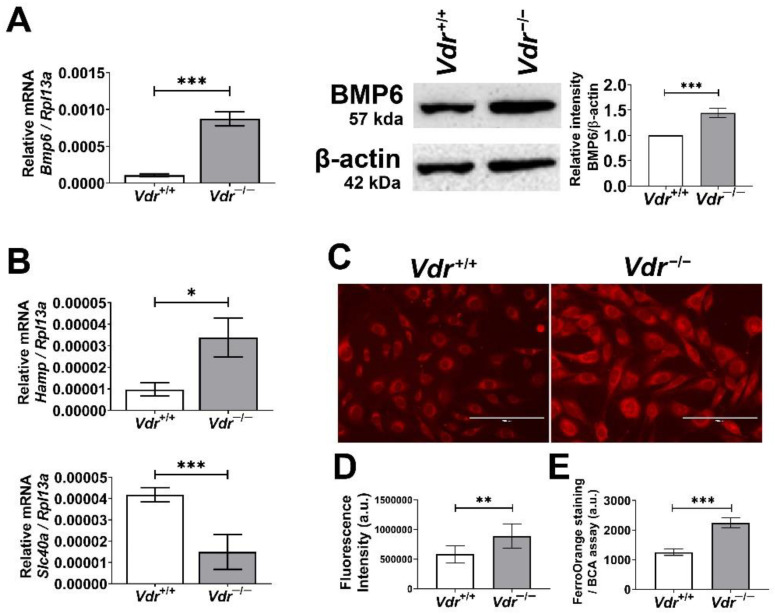
Alterations in expression of genes involved in iron homeostasis and intracellular iron levels in retinal EC. BMP levels were determined by qPCR analysis (**A**) and Western blot analysis (**B**) (n = 3, *** *p* < 0.001). (**B**) mRNA levels of Hepcidin (Hamp) and Ferroportin (Slc40a) in *Vdr*^+/+^ and *Vdr*^−/−^ retinal EC were determined by qPCR analysis (n = 3, * *p* < 0.05, *** *p* < 0.001). (**C**) Intracellular iron levels in retinal EC were analyzed by staining cells with FerroOrange probe and (**D**) measuring fluorescence intensities of FerroOrange staining (Scale bars = 200 µm, n = 6–7, ** *p* < 0.01). (**E**) Alternatively, cells were plated in 96-well plates for ferroOrange staining and fluorescence intensities were measured using a plate reader and normalized to total protein levels (n = 3, *** *p* < 0.001).

**Table 1 cells-12-00335-t001:** The mouse primers used in qPCR analysis.

Gene	Forward (5′ to 3′)	Reverse (5′ to 3′)
Vdr	GGCTTCCACTTCAACGCTATG	TGCTCCGCCTGAAGAAACC
Mcp-1	GTCTGTGCTGACCCCAAGAAG	TGGTTCCGATCCAGGTTTTTA
Il-6	CAACCACGGCCTTCCCTACT	TTGGGAGTGGTATCCTCTGTGA
Il-33	GGTGAACATGAGTCCCATCA	CGTCACCCCTTTGAAGCTC
St2	GGACCATCAAGTGGAGGGAA	GCACTGGCATTTGGTACCTC
Cxcl1	ACAGGGGCGCCTATCGCCAA	CGGTTTGGGTGCAGTGGGGC
Cxcl2	CCCTTGGACATTTTATGTCTTCC	GACACGAAAAGGCATGACAA
Icam1	GCCATAAAACTCAAGGGACAA	GGCTGAGGGTAAATGCTGTC
Vcam1	TCGCGGTCTTGGGAGCCTCA	TGACTCGCAGCCCGTAGTGC
Ager	GTCACAGAAACCGGCGAT	TACTACTCCCAGGCCTCCC
F3	AAGTGCTTCTCGACCACAGA	TGGGACAGAGAGGACCTTTG
Fas	TGCTTGCTGGCTCACAGTTA	TATCAGTTTCACGAACCCGC
Vegf	GGAGAGCAGAAGTCCCATGA	ACTCCAGGGCTTCATCGTTA
Stat3	ACCAACATCCTGGTGTCTCC	CACTACCTGGGTCGGCTTC
Plau	CGATTCTGGAGGACCGCTTA	GACACCGGGCTTGTTTTTCT
Nos2	GGCAGCCTGTGAGACCTTTG	CATTGGAAGTGAAGCGTTTCG
Piezo1	TCCTCTTCCTCATCGCCATC	AGGGTGACGGTGACATCAAT
Piezo2	CCCATTTCTGACTGAGCTGC	ATGTGCGCGTAAATGTCCTC
Bmp6	GTGACACCGCAGCACAAC	TCGTAAGGGCCGTCTCTG
Hamp	GCATCTTCTGCTGTAAATGCTG	TGGCTCTAGGCTATGTTTTGC
Slc40a	ATGTGAACAAGAGCCCACCT	CCCATCCATCTCGGAAAGT
Rpl13a	TCTCAAGGTTGTTCGGCTGAA	GCCAGACGCCCCAGGTA

## Data Availability

All the data presented here are included in the manuscript. Further inquiries should be directed to the corresponding author.

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
