# Peer review of "Vitamin D Receptor Expression Limits the Angiogenic and Inflammatory Properties of Retinal Endothelial Cells"

_cells, 2023, doi:10.3390/cells12020335_

Round 1
Reviewer 1 Report
1) Shorten the introduction especially Line 71 to 86. They read like a long summary of your previous studies. Also, manuscript will be benefited if you can make a separate paragraph to list what is unknown in the field and how is this study trying to fill the gap.
2) Why was no ‘negative sorting’ done while EC isolation from retina?
3) Since VDR plays several important roles in EC homeostasis; therefore, I am wondering was there any difference in cell viability during isolation?
4) Authors should include a negative control such as CD45 in Figure 1D.
5) Provide scale bar for Fig 2A, 3C, 7B
6) Is there a specific reason that p120 catenin staining not done in Figure 2A?
7) ZO-1 representative images especially staining is not convincing of an increase in ZO-1. Therefore, authors should repeat ZO-1 western blot and staining.
8) Please be consistent with type of graphs within figures. Some panels are representative as bar graphs whereas some contains individual data points.
9) In Fig 3, if possible, show phallodinn and vinculin (red) separately because due to merging vinculin is showing as yellow or you can change figure legend.
10) In Fig 5, no evidence to show equal loading for western blot done on culture media. Authors can use ponceau staining as loading control.
11) In Fig 6, why was ICAM-1 and VCAM-1 levels not tested as they are major target of NF-kB signaling?
12) Why were intracellular ROS or GSSG/GSH ratio analyzed in Figure 8.
13) Regarding line 559 to 571- your analysis was not exclusive on IL-33; therefore, why is discussion focused on IL-33 alone. You can include discussion about overall inflammation.
14) Line 606 reads as “we also note increased oxidative stress”. However, you only tested susceptibility to oxidative stress via looking at cell viability. Therefore, you should change your conclusion.
15) No conclusion provided at the end of manuscript. It will be helpful if you can write an overall summary of your results and try to bring all your experiments together.
Author Response
1) Shorten the introduction especially Line 71 to 86. They read like a long summary of your previous studies. Also, manuscript will be benefited if you can make a separate paragraph to list what is unknown in the field and how is this study trying to fill the gap. This is revised as suggested.
2) Why was no ‘negative sorting’ done while EC isolation from retina? We use magnetic beads coated with antibody to CD31 to pull out EC. The specificity was minimally provided by CD31 and VE-cad positivity by FACS as shown here. During the initial isolation, we also look for B4-lectin positive staining (a specific lectin for mouse EC), as well as for lack of Pdgfrb and SMA (pericyte/smooth muscle cell markers). This is now included in Figure 1 legend as not shown.
3) Since VDR plays several important roles in EC homeostasis; therefore, I am wondering was there any difference in cell viability during isolation? A shown in Figure 8 the basal cell viability between Vdr+/+ and Vdr-/- EC is the same. We also do not see any significant differences in development of retinal vasculature in Vdr+/+ and Vdr-/- mice (20).
4) Authors should include a negative control such as CD45 in Figure 1D. Please see response to question 2.
5) Provide scale bar for Fig 2A, 3C, 7B. These panels have scale bar and is also indicated in the figure legend.
6) Is there a specific reason that p120 catenin staining not done in Figure 2A? Generally, when one looks at adherens junction organization, P120 staining is not informative by localization but it is important to assess its level by Western for potential changes in protein levels as done here.
7) ZO-1 representative images especially staining is not convincing of an increase in ZO-1. Therefore, authors should repeat ZO-1 western blot and staining. This is now done, and a new figure is provided.
8) Please be consistent with type of graphs within figures. Some panels are representative as bar graphs whereas some contains individual data points. We apologize for this oversight. We have now addressed this, and all graphs are show as bar graph.
9) In Fig 3, if possible, show phalloidin and vinculin (red) separately because due to merging vinculin is showing as yellow or you can change figure legend. This is now done as requested.
10) In Fig 5, no evidence to show equal loading for western blot done on culture media. Authors can use ponceau staining as loading control. For these assays, we normally plate same number of cells and use the total cell lysate actin staining to control for loading. We cannot use actin with condition medium or other markers since levels are low to control for loading. Ponceau staining of the membrane is a good suggestion and we have done that in the past for comparison of the two method of normalization and results were similar. That is why we do not routinely do ponceau staining.
11) In Fig 6, why was ICAM-1 and VCAM-1 levels not tested as they are major target of NF-kB signaling? This is now done and added to Figure 6.
12) Why were intracellular ROS or GSSG/GSH ratio analyzed in Figure 8. Since basal viability was not different, we wanted to simply see if Vdr-/- cells are more sensitive when challenged with H2O2, a reactive oxygen species. This is generally consistent with diminished antioxidant capacity of the cells.
13) Regarding line 559 to 571- your analysis was not exclusive on IL-33; therefore, why is discussion focused on IL-33 alone. You can include discussion about overall inflammation. This is fixed as suggested.
14) Line 606 reads as “we also note increased oxidative stress”. However, you only tested susceptibility to oxidative stress via looking at cell viability. Therefore, you should change your conclusion. This is now corrected as suggested.
15) No conclusion provided at the end of manuscript. It will be helpful if you can write an overall summary of your results and try to bring all your experiments together. This is fixed as suggested.
Reviewer 2 Report
Vitamin D receptor (VDR) is a nuclear receptor whose engagement by vitamin D impacts the expression of many genes with important roles in regulation of angiogenesis and inflammation. In this study, using retinal endothelial cell (EC) isolated from wild-type (Vdr+/+) and Vdr-deficient (Vdr−/−) mice, Song et al has proved that VDR expression plays a fundamental role in maintaining the proper angiogenic and inflammatory state of retinal EC. The lack of VDR expression had a significant impact on endothelial cell-cell and cell-matrix interactions. They also exhibited increased expression of inflammatory signals driven by the activation of STAT1 and NF-κB pathways and were more sensitive to oxidative challenge.
Major:
The major concern from this reviewer is the gross phenotype of the Vdr-deficient (Vdr−/−) mice, especially those related to vision formation. The vitamin D endocrine system is suggested with a more widespread function. The VDR is nearly ubiquitously expressed. Is there any specific phenotype or predisposition to any retina related disease for the Vdr−/− mice? If this question can not be answered first, the biological research from isolated retinal EC does not make sense.
Minor:
1. The immunofluorescence staining of this submission needs to be improved generally, the staining results are not convincing, for example, Figure 2A.
2. Figure 3B, the images without any staining or at such a lower lower magnification are hard to be matched with the statistical results of cell migration.
3. Figure 7B, are the immunofluorescence staining of P65 and p-P65 specific? why is the p-p65 expression in Vdr−/− mice increased greatly in the cytoplasm of ECs?
Author Response
Vitamin D receptor (VDR) is a nuclear receptor whose engagement by vitamin D impacts the expression of many genes with important roles in regulation of angiogenesis and inflammation. In this study, using retinal endothelial cell (EC) isolated from wild-type (Vdr+/+) and Vdr-deficient (Vdr−/−) mice, Song et al has proved that VDR expression plays a fundamental role in maintaining the proper angiogenic and inflammatory state of retinal EC. The lack of VDR expression had a significant impact on endothelial cell-cell and cell-matrix interactions. They also exhibited increased expression of inflammatory signals driven by the activation of STAT1 and NF-κB pathways and were more sensitive to oxidative challenge.
Major:
The major concern from this reviewer is the gross phenotype of the Vdr-deficient (Vdr−/−) mice, especially those related to vision formation. The vitamin D endocrine system is suggested with a more widespread function. The VDR is nearly ubiquitously expressed. Is there any specific phenotype or predisposition to any retina related disease for the Vdr−/− mice? If this question can not be answered first, the biological research from isolated retinal EC does not make sense.
Our interest in Vitamin D came from its potent anti-angiogenic activity. We previously showed calcitriol, the active form of vitamin D, is a potent inhibitor ocular neovascularization in the oxygen-induced ischemic retinopathy, a model of retinopathy of prematurity and a major cause of vision impairment in premature babies. These studies thus suggested administration vitamin D may have therapeutic potential in babies born prematurely, which needs further investigation.
We became interested in how vitamin D inhibits angiogenesis. To start addressing this question, we wished to determine the cell autonomous impact of vitamin D/Vdr signaling in regulation of angiogenesis. We have shown that inhibition of angiogenesis in our preclinical model is Vdr dependent, since Vdr-deficient mice do not respond effectively to mitigation of angiogenesis by calcitriol. I should also point out that the key phenotype of vitamin D deficiency noted in Vdr-/- mice is not manifested until after weaning. To keep these mice healthy after weaning special diet is needed. Looking at the development of retinal vasculature or neovascularization we did not see significant differences since these all occur before weaning.
Our cell autonomous study of Vdr action has been very informative as previously reported for retinal PC and here for retinal EC. These studies have provided significant insight to potential function of Vdr in vascular development, neovascularization, and vascular cell homeostasis.
Minor:
- The immunofluorescence staining of this submission needs to be improved generally, the staining results are not convincing, for example, Figure 2A. This is fixed.
- Figure 3B, the images without any staining or at such a lower magnification are hard to be matched with the statistical results of cell migration. The cells are visualized by blue H/E staining of the filters. The magnification x400 and shows the blue stained cells which are more abundant in Vdr+/+ panel. We now added some arrows pointing to the cells that have migrated through the filter and stock to the bottom of the filter for clarification.
- Figure 7B, are the immunofluorescence staining of P65 and p-P65 specific? why is the p-p65 expression in Vdr−/−mice increased greatly in the cytoplasm of EC? This could be in turn due to morphology changes noted in these cells. The bigger cells appear having more cytoplasmic staining.